

# Reproductive benefits and reduced investment in parental care behavior associated with reproductive groups of males in *Abudefduf troschelii*

Mariana Solís-Mendoza[1], Omar Chassin-Noria[1], Carlos Levi Pérez Hernández[2] and Luis Mendoza-Cuenca[2]

[1] Facultad de Biología/Centro Multidisciplinario de Estudios en Biotecnología, Universidad Michoacana de San Nicolás de Hidalgo, Morelia, Michoacán, Mexico
[2] Laboratorio de Ecología de la Conducta/Facultad de Biología, Universidad Michoacana de San Nicolás de Hidalgo, Morelia, Michoacán, Mexico

Corresponding author
Luis Mendoza-Cuenca,
lfmendoza@umich.mx

## ABSTRACT

Fishes of the family Pomacentridae present a wide diversity of mating systems, ranging from polygyny to promiscuity and from individual territorial defense to the establishment of reproductive colonies of males. The damselfish species *Abudefduf troschelii* has a reproductive colony mating system, in which males temporarily aggregate in reproductive areas to court and attract females. Males defend an individual territory where they receive eggs and perform paternal care behaviors for their offspring. The present study evaluated the advantages of the colonial mating system in *A. troschelii*. During an entire reproductive period, in a breeding colony within a rocky reef, we located, marked, geo-referenced, and measured the distances between the territories of all males. We quantified the variance among males in their patterns of paternal care investment, eggs acquired, hatching success, reproductive success, body size, and changes in body coloration. We found that males spatially distributed their nests in groups or independently (*i.e.*, solitary nests). Nesting groups are formed by larger males that show intense nuptial coloration during the entire receptivity period. They are located centrally to the colony and consist of three to six males whose territories overlap. In contrast, small solitary males that fail to acquire or maintain nuptial coloration during the receptivity period establish their nests peripherally to the colony, away from the territories of other males. Our results highlight that the reproductive benefits of colonial nesting are unequal for males, as the spatial distribution of nests within the colony determines the reproductive success of males. Group nesting confers the highest reproductive benefits to males regarding eggs obtained, hatching success, and relative fitness and also enables males to reduce their parental investment in brood care behaviors. The preference of females for oviposition could be associated with greater intrasexual competitiveness, defense ability, body condition, or experience of group-nesting males located at the center of the colony or because their progeny will have a lower probability of predation than they would in solitary nests males.

# INTRODUCTION

The spatial and temporal distribution of potential mates in a habitat can determine factors of great importance on the fitness of individuals, including mate choice and variance in reproductive success among individuals of a population (*Emlen & Oring, 1977*; *Shuster & Wade, 2003*; *DuVal, Vanderbilt & M'Gonigle, 2018*). Mating systems (MS) that involve the grouping of males performing courtship displays in adjacent areas to attract females are common in nature and perhaps the best studied example of this is that of the Lek mating system, where males form groups during the mating season, in such leks they court and are visited by the females for the sole purpose of mating (*Höglund & Alatalo, 1995*).

The Lek mating system has been described in freshwater and marine fishes (*McKaye, 1983*; *McKaye, Louda & Stauffer Jr, 1990*; *Jan, 1991*; *Kellogg et al., 1995*; *Genner et al., 2008*). However, in most marine species, the males defend individual territories and receive eggs from the females within these groups and are then left with exclusive responsibility for providing parental care to the progeny. Paternal care represents an extra contribution of the males (in addition to their gametes) and comes into conflict with the definition of Lek (*Höglund & Alatalo, 1995*). In this context, different authors have opted instead to refer to them as "groups", "Lek-like" MS (*Thresher, 1984*), "colonial nesting" (*Tyler III, 1995*) or "colonial breeding" (*Schütz et al., 2016*).

Reproductive group behavior has evolved in multiple species, frequently associated with an increased individual fitness (*Krause et al., 2002*). However, few studies have provided empirical data on how ecological factors (*e.g.*, predation) or life history traits (*e.g.*, body size) may shape the distribution of males within breeding colonies or the fitness benefits of males practicing colonial nesting (*Tyler III, 1995*; *Young et al., 2009*; *Rueger et al., 2021*). Some authors have empirically shown that models of lek evolution are applicable to explain colonial breeding models (*Schütz et al., 2016*). The possible theoretical advantages proposed for the evolution of colonial breeding include the indirect benefits of living in groups when closely related individuals live together, increasing the inclusive fitness of close relatives (*Dominey, 1981*; *Gross & MacMillan, 1981*; *Tyler III, 1995*). In addition, it has been suggested that females prefer to appraise males in groups (*Bradbury, 1981*; *Fletcher & Miller, 2006*). For this reason, male aggregations during the breeding seasons have the benefit of increasing mate attraction and reproductive success per capita (*Gross & MacMillan, 1981*; *Côté & Gross, 1993*; *Tyler III, 1995*). Within groups, females appear to prefer males with centrally located nests compared to the peripheral males, which in this context could act as a barrier against the intrusion of predators (*Foster, 1989*), and reduce egg loss (*Gross & MacMillan, 1981*; *Tyler III, 1992*; *Tyler III, 1995*). Furthermore, reproductive groups could improve females' efficiency in evaluating phenotypic traits related to males' fitness or quality of parental care, such as body size, aggressiveness, or territory size (*Coté & Hunte, 1989*). In some species, the females also use male coloration

as an estimator of hierarchy, aggressiveness, quality of courtship (*Seehausen & van Alphen, 1998*), parasitism level, quality of immune response (*Møller, Christe & Lux, 1999*; *Clotfelter, Ardia & McGraw, 2007*) their capacity to defend the progeny and the quality of the territory (*Maan et al., 2004*; *Dijkstra, Vander Zee & Groothuis, 2008*; *Genner et al., 2008*). Even if males use fabricated structures as extended phenotype signals, this could also be used for females to check male quality economically (*Schaedelin & Taborsky, 2006*; *Schaedelin & Taborsky, 2010*; *Mitchell, Ocana & Taborsky, 2014*).

The costs of colonial nesting include increased conspicuity as group size increases and, therefore, an increased probability of predation for males and their progeny (*Tyler III, 1995*). Moreover, the closer proximity between nests increases the likelihood of parasite transmission, intrasexual competition, reproductive interference during spawning and the risk of paternity loss (*Alexander, 1974*; *Brown & Brown, 1986*; *Tyler III, 1995*; *Guillen-Parra et al., 2020*). Because of the cost-benefit trade-off of colonial nesting behavior, selective pressures to optimize nest distances to maximize male reproductive success might be expected (*Krause et al., 2002*).

Reef fishes of the genus *Abudefduf* (family: Pomacentridae) engage in multiple reproductive bouts throughout the year (*Fishelson, 1970*; *Foster, 1987*; *Robertson, Petersen & Brawn, 1990*). In the beginning, males acquire a darker-bluish nuptial coloration (*Bessa, Dias & de Souza, 2006*), establish and defend a territory, prepare the nest area, court females and receive clutches of eggs from one or several mates (*Tyler III, 1995*; *Lobel et al., 2019*; *Guillen-Parra et al., 2020*). Males receive clutches for three days (*Foster, 1987*; *Tyler III, 1995*). Once oviposited, the eggs remain under the protection of the male until the larvae hatch (*Fishelson, 1970*; *Foster, 1987*). Paternal care lasts 6 to 11 days; it is costly since males suspend foraging (*e.g.*, as in *A. abdominalis*, *Tyler III, 1995* to remain in their territory, oxygenating and defending the eggs against possible predation (*Fishelson, 1970*; *Foster, 1987*; *Foster, 1989*; *Pérez-Hernández, 2018*). Fishes of this genus present different reproductive mating systems, including the territorial defense of solitary or group nests within breeding colonies (*Foster, 1989*; *Tyler III, 1992*; *Tyler III, 1995*). The number of individuals that form a group can vary, as well as the distance at which the closest male is found (*Foster, 1989*; *Tyler III, 1995*). Alternative reproductive tactics have also been described, such as the presence of sneaking males that try to fertilize the eggs under the guard of other males during oviposition (*Goldschmidt, Foster & Sevenster, 1992*; *Dougherty et al., 2022*). For *Abudefduf* species, reported percentages of paternity theft range from 5% (in 15% of nests) in *A. sordidus* (*Lobel et al., 2019*) to about 50% in *A. troschelii* (*Guillen-Parra et al., 2020*). However, in other fish groups, the success rate of parasitic males (one or more per nest) can reach up to 78% (*Wirtz et al., 2014*).

*Abudefduf troschelii* is a reef fish with diurnal habits. It moves in shoals on mid-depth water without establishing territories and feeds mainly on plankton (*Foster, 1987*; *Aguilar-Medrano et al., 2011*). At the beginning of the mating season, males separate from the conspecific school. They display a conspicuous coloration change (*i.e.,* sexual dichromatism) from yellow darkening until reaching a bluish tone (*Tyler III, 1995*), which has been suggested as nuptial coloration (*e.g.*, *A. saxatilis*, *Bessa, Dias & de Souza, 2006*). Breeding colonies in this species are formed by groups of males nesting simultaneously

(from 1 to over 50 males) with a very short distance to the nearest neighbor's nest (0.48 ± 0.25 m; *Foster, 1989*). Territorial males court females and, if they receive egg clutches, are exclusively responsible for the parental care of the immature until the pelagic larvae hatch (*Fishelson, 1970*; *Foster, 1987*; *Pérez-Hernández, 2018*). For *A. troschelii* males, the reproductive period lasts about 7–11 days and includes a brief three-day spawning period (3.67 d, SD = 1.29; *Foster, 1987*), during which they court females and receive egg clutches from one or more females (*Guillen-Parra et al., 2020*). After this period, males invest all their parental effort in guarding their territories and the progeny in them until larval hatching typically occurs four or five days after spawning (*Fishelson, 1970*; *Foster, 1987*).

In this context, this work aims to evaluate the benefits of colonial nesting and the traits contributing to its evolution in damselfishes. Therefore, we studied nesting and parental care patterns in a male breeding colony of the damselfish *Abudefduf troschelii* to evaluate: (1) whether the spatial distribution of nests is random or shows aggregation patterns within the colony; (2) the effect of phenotypic characteristics of males on the spatial distribution patterns of their nests within the colony; (3) if male nesting decisions determine the reproductive benefits (*e.g.*, reproductive success, parental investment) associated with colonial breeding.

## MATERIALS & METHODS

### Study site

The study was conducted between September 19th and October 4th, 2016, in an *A. troschelii* breeding colony located in a homogeneous and continuous rocky reef in the Bay of La Paz, Baja California Sur, Mexico (24°9′22.32″N, 110°19′28.00″W), at a water depth of between 1 and 5 m. We monitored the area daily starting 5 days before the full moon (*i.e.*, non-nesting period, *Tyler III, 1995*) and until males began nest preparation behavior (September 25–26). All of the initiated territories (*i.e.*, areas of the reef cleaned by a male) were marked with a numbered rock, geo-referenced and the inter-nest euclidean distances were measured with a 100 m measuring tape to determine the male spatial nesting patterns and generate a detailed map with the spatial distribution of all nests in the colony (Fig. 1). The distribution of the territories was used to determine whether the males of *A. troschelii* in this population establish group territories or conduct solitary nesting (see *Foster, 1989*; *Tyler III, 1995*): (1) Nesting groups are formed by the territories of three or more individuals that simultaneously initiate nest preparation and egg reception. The maximum distance to the center of the nearest neighbor's nest is less than 1 m (see *Foster, 1989*; *Tyler III, 1995*, and the areas defended by males overlapped (see below for the male's maximum attack distance definition). (2) Solitary nesting is performed by males who initiate a nest at a distance larger than 2.5 m to the center of the nearest neighbor's nest. Therefore their defended areas do not overlap with those of other males in the breeding colony.

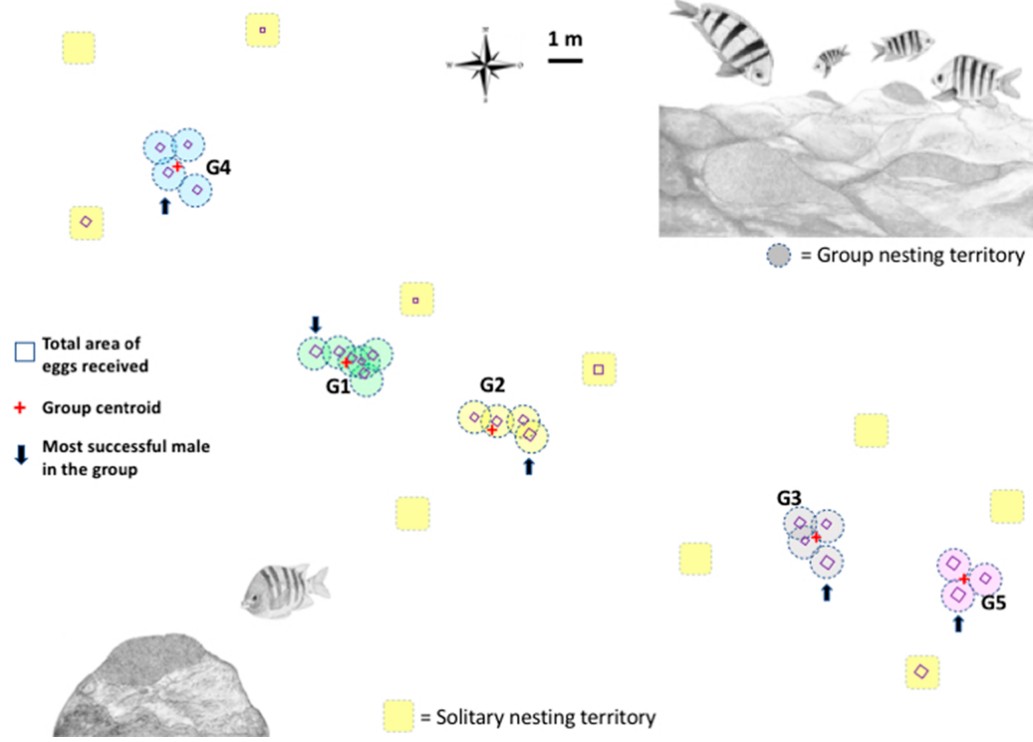

**Figure 1** **Map of the breeding colony of *Abudefduf troschelii*.** The geo-referenced scaled location of male nests with a group and solitary nesting is shown. The total number of eggs each male receives is shown as squares to scale. We use the average size of all the territories in the population (0.95 m) to illustrate the size of the territory on the map.

## Observations of reproductive behavior

All territories were monitored using daily 15-minute video recordings made over the entire reproductive period (*i.e.,* from spawning until larval hatching 4–5 days). These recordings were made with waterproof cameras (GoPro Hero 4 Silver), located 50 cm from the center of each territory, to avoid agonistic behavior of the male towards the camera and a loss of detail in the recording associated with the water depth and focus of the cameras. Each video featured a size reference placed within the frame to estimate the male's size and territory (see below). The daily order of video recording was randomized between 9:00 and 14:00 hrs. We used these videos to quantify the time spent on parental care behaviors and the reproductive success of each individual. The videos of each male (*i.e.,* one per nesting day) were viewed second by second to quantify the time (in minutes) spent by each male on the different parental care behaviors. As an estimator of the investment in parental care performed by males, we used the time (in minutes) allocated by each male to the main parental care behaviors previously reported at the study site by *Pérez-Hernández (2018)*, both per nesting day and per entire nesting period (sum of seconds allocated daily). We also estimated the proportion (*i.e.,* percentage) of the total time allocated by each male to these behaviors. Blinded methods were used when all parental care behaviors were quantified from all video recordings to minimize observer bias (the observers knew neither

the individual identity, the order of the videos, nor the strategy of each male). The patterns into which we organized the parental care behavior were the following:

**Chasing:** The male chases away an intruder that enters his territory.

**Oxygenation of eggs:** Propulsion of water towards the eggs by fanning the pectoral fins or propelling water using the mouth.

**Guarding:** The male is located over the nest (at an approximate distance of no more than 20 cm) while regularly changing body orientation to cover a 360° field of view and toward the clutches and the surface.

The time the male leaves the nest area was recorded (*i.e.,* beyond the video frame), with no association to any parental care behavior (*e.g.*, chasing). Therefore, we label this time as absence since we have no direct evidence of males' behavior (*e.g.*, foraging, courtship) when they are absent from the nest.

To estimate male size from the videos, we took several photographs of the complete individual besides the size reference. From these images, we measured each male's total length (TL: the distance from the mouth to the tip of the longest lobule of the caudal fin). For all videos, we measured the distance from the nest's center to the maximum point at which the male chased an intruder and used it as the male's maximum attack distance (MDA) radius. If the male left the video frame during a chase, the distance from the nest's center to the last observable point in the video frame (*i.e.,* 98 cm) was considered the MDA radius. The MDA diameter was used to estimate the territory size for each male. Finally, in all videos, we recorded whether the male presented nuptial coloration.

In order to estimate the number of eggs received by each male, daily photographs were taken twice a day (9:00 and 16:00 hrs) from the day the males received their first egg clutch until the hatching of all larvae. The camera (Olympus TG-5 camera) was positioned parallel 30 cm from the egg clutch (*i.e.,* nest), and a size scale was placed next to the eggs for size reference. If the male nest consisted of egg clutches located on separate rocks (*i.e.,* sections), a photograph was taken in each section. The program ImageJ ver. 1.53. was used to estimate the area occupied by eggs, and daily variations in this value were recorded (*Guillen-Parra et al., 2020*). We also employed *in situ* macro-photography of a one cm$^2$ area (with an Olympus TG-5 camera), to quantify twice a day (9:00 and 16:00 hrs) the number of eggs per cm$^2$ in each nest. We take one macro-photograph at the center of the egg clutch and two more at 10 cm from (in each section of the nest). These macro-photographs allowed us to make an estimate of egg density per cm$^2$ and a rough estimate of the number of eggs received by the male (total area of egg clutch × egg density). In addition, macro-photographs at the center of the nest allowed us to document egg development time; from spawning to hatching (Fig. 2), which confirmed that larvae hatch on the 4th or 5th day after egg oviposition (at night). Following *Foster (1989)*, we considered the hatching success of each male as the total area of the egg clutch present in the last photograph of the afternoon of the fourth and fifth nesting day that is no longer present in the first photograph of the following morning (penultimate and last nesting day).

To evaluate the effect of the reproductive strategies on the male's reproductive success, we used three independent fitness estimators: (1) total area of eggs received; (2) hatching

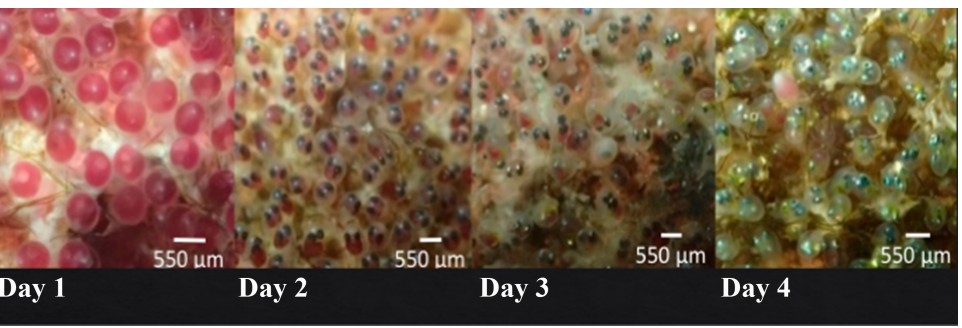

**Figure 2** **Images of the development of *A. troschelii.* eggs per day.** From the day of oviposition (day 1) to the day of hatching (days 4–5).

success, the proportion of the total area of eggs received from which larvae hatched (area with hatched eggs/total area of received eggs); (3) relative hatching success (total area of eggs hatched from each male/mean area of eggs hatched from the total males of the population) (*Gillespie, 2004*).

This study was carried out in strict accordance with the recommendations of the "Regulation of the Institutional Ethics Committee" of the Universidad Michoacana de San Nicolás de Hidalgo.

## Data analysis

The nests of all males in the breeding colony were spatially mapped in a two-dimensional space (*x*–*y* coordinates point pattern) in order to determine whether, within the breeding colony, males distribute their nests in a continuous (*i.e.,* complete spatial randomness, CSR) or clustered manner. We performed a Ripley's K function, using the Monte Carlo method and 1,000 simulated CSR, and corrected for any border effect in the spatstat package in R software.

In order to obtain a more detailed description of the contribution of male quality, parental care and the reproductive strategy on the reproductive success of the males, and to compare the effect in terms of reproductive success of the nesting strategies practiced by the males of *A. troschelii* in the study population, three independent generalized linear mixed models (GLMM) were generated using the identity of the groups as a random variable and a Normal distribution of errors. These models used the following estimators of fitness as response variables: generalized linear mixed model of the total area occupied by eggs ($GLMM_{TA}$), generalized linear mixed model of hatching success ($GLMM_{HS}$) and generalized linear mixed model of relative hatching success ($GLMM_{RHS}$). Reproductive strategy (*i.e.,* grouped males or solitary nesting), investment in behaviors of parental care (*i.e.,* time assigned to each of the behaviors of parental care), absence of the nest and male size were used as independent variables. Because all three response variables contain zero values, GLMMs were performed in R software using the glmmTMB function of the glmmTMB package (*Brooks et al., 2017*), which allows the modeling of zero-inflated continuous data. In addition, for the data analysis, the total time of nesting by the males

was divided into two periods: (1) receptivity period (from the day males concluded nest cleaning and received their first eggs, until the last day on which they performed courtship behavior and received clutches of eggs); and (2) hatching period (from the day after the end of receptivity to eclosion of the last larvae). In the case of GLMM $_{TA}$, only the receptivity period was used for the investment of parental care behaviors, since these could influence mate selection and oviposition preference on the part of the females. Non-significant explanatory variables in each full model were backward eliminated to simplify the model.

We compare male size, durations, and frequencies of behavioral patterns of parental care and absence of the nest between group nesting and solitary nesting males of *A. troschelii* performing a series of generalized linear mixed models (GLMM) using the identity of the groups as a random variable. We use a normal distribution for the male size response variable and a Poisson distribution of errors for all behavioral response variables. The analyses were performed in R software.

Due to conspicuous egg coloration in *A. troschelii*, females could visually quantify the total number of eggs in the nest of each male and prefer nests with more eggs. Bias in female oviposition preferences for traits such as the presence, size, and color of egg clutches has been previously reported in teleost fish species with exclusive paternal care, including *A. sexfasciatus* and *A. luridus* (*Afonso & Santos, 2005*; *Goldberg et al., 2020*). We evaluated whether females thus biased their oviposition preferences across receptivity days. We performed a linear mixed-effects model (GLMM $_{EP}$) using the total number of eggs in each nest per day as the response variable (during the 3 days of receptivity); as fixed effects, we considered the nesting day, the number of eggs gained and the interaction day*eggs gained, and as random effects the intercept and the nesting day within each male. Since many solitary males did not receive eggs, we used only group-nesting males.

From the distribution of the groups, we evaluated whether the relative position of each male's nest influences his reproductive success (*e.g.*, female preference for centrally located nests). We used the center of each nest in a group to draw an irregular polygon, calculate its centroid and use it as the group's center. Next, we measured the distance between this point to the center of all the group nests and the center of the nearest solitary nests. Using these distances we evaluated: (1) the relationship between male size and the location of their nesting territories in the breeding colony; (2) the effect on individual success of the position of the territory of each male relative to the centroid of its group (for group nesting males) or to the centroid of the nearest group (for solitary nesting males). For this, because the relationship between the variables is not linear we used Spearman rank correlations, with distance from the center of the territory of each male to the centroid of the group as an independent variable (in the case of solitary nesting males, the distance to the centroid of the nearest group was used), male size (*i.e.,* total length) and the three fitness estimators as dependent variables. Furthermore, to evaluate whether there is an effect of the position of the territory of each male relative to the center of the territory of the most successful male in each group (*i.e.*, the male with the largest area of nesting territory occupied by eggs), we repeated the Spearman rank correlation analysis described above, but using the distance from the center of the territory of each male to the center of the territory of the most

successful male in each group as the independent variable. For all analyses, Bonferroni corrections were applied to correct the no independence between variables.

## RESULTS

More than 2,300 min of video were analyzed, corresponding to the behaviors of parental care in 31 custodian males that could be observed daily over the entire course of their reproductive period (mean = 3 days, range 1–6). Two types of nest spatial distribution were observed: (1) males exhibiting group nesting behavior ($n = 21$ individuals), with an average distance to the nearest neighbor of 70.8 cm ($\pm$ 14.4), which presented nuptial coloration and maintained it during the 3 days of receptivity; and (2) those practicing solitary nesting SN ($n = 10$ individuals), with an average distance to the nearest neighbor of 296. 4 cm ($\pm$ 66.3), of these, 50% showed nuptial coloration and only 10% maintained it for at least the first 2 days of receptivity; Fig. 1. The 21 group-nesting males were found in five independent groups (G1 = 6 inds., G2 = 4 inds., G3 = 4 inds., G4 = 4 inds., G5 = 3 inds.). The mean length of all males was 13.54 cm (range 10.98–17.3 cm). The Kolmogorov–Smirnov spatial test of CSR (Ripley's K function) showed that nests were not continuously distributed ($D = 0.678$, $p < 0.001$). Ripley's K function strongly suggests that the highest proportion of *A. troschelii* males nest near their neighbors, closer than would be expected by chance (Fig. S2).

The generalized linear models revealed that the male reproductive strategies showed significant differences in the three estimators of fitness considered. The GLMM $_{TA}$ to evaluate differences in mating success, showed significant effects of the strategy, male size and guarding parental care (Table 1), higher egg production success was observed in males nesting in groups (Table 2, Fig. 3). The GLMM $_{HS}$ revealed significant effects of the strategy, guarding and absence behavior during the final day of the nesting period (Table 1), with group-nesting males having the highest hatching success (Table 2, Fig. 3). The GLMM $_{RHS}$ to evaluate the differences in relative hatching success revealed significant effects of the strategy and guarding parental care behavior (Table 1), the highest relative success is achieved by group-nesting males (Table 2, Fig. 3). The general patterns of the parental care behaviors of all males that received eggs show that the males of the population assign most of their time to guarding behavior (*ca.* 70%), although they also spend some time absent from the nest (*ca.* 25%). The mean area occupied by eggs per nest was 416.93 cm$^2$ (range = 0–969.02 cm$^2$), the mean area of eggs under guard by the males at the end of the hatching period was 139.07 cm$^2$ (range = 0–672.73 cm$^2$), while the mean hatching success of the males of the population was 27% (range = 0–91%).

Results of GLMM analyses to compare morphological traits, parental care behaviors, and fitness estimators between group nesting and solitary nesting males showed that group-nesting males spent significantly a higher percentage of parental care time on guarding behavior (Estimate = 24.61, S.E. = 6.26, $t$-value = 3.93, $p = 0.00048$), they lost a lower percentage of eggs (Estimate = −6.74, S.E. = 0.56, z-value = 11.88, $p < 0.001$) and spent a higher percentage of time absent from the nest (Estimate = −5.64, S.E. = 1.29, $t$-value = −4.46, $p < 0.001$). However, no significant differences were observed for male

**Table 1  Comparison among the patterns of parental care between the nesting behavior presented by males of *A. troschelii*.**

| | | Model GLMM$_{TA}$ | | |
|---|---|---|---|---|
| **Parameter** | **Estimate** | **S.E.** | **Z value** | **P** |
| Strategy | −246.010 | 88.480 | −2.780 | 0.0054 |
| Male size | 51.784 | 20.833 | 2.486 | 0.0129 |
| Guarding | −2.596 | 2.147 | −1.209 | 0.2265 |
| | | Model GLMM$_{HS}$ | | |
| **Parameter** | **Estimate** | **S.E.** | **Z value** | **P** |
| Strategy | −0.3758 | 0.1050 | −3.550 | 0.00038 |
| Absence | −0.0097 | 0.0044 | −2.171 | 0.02996 |
| Guarding | −0.0101 | 0.0039 | −2.550 | 0.0107 |
| | | Model GLMM$_{RHS}$ | | |
| **Parameter** | **Estimate** | **S.E.** | **Z value** | **P** |
| Strategy | −0.6881 | 0.3402 | −2.012 | 0.0442 |
| Guarding | −0.0153 | 0.0079 | −1.926 | 0.0541 |

**Table 2  Male size, patterns of parental care and estimators of fitness between the nesting behavior presented by males of *A. troschelii*.** Significant differences among nesting strategies are shown.

| Male trait | Group nesting (mean ± S.D.) | Solitary nesting (mean ±S.D.) | GLMM p-value |
|---|---|---|---|
| Size of male (TL in cm) | 13.60 ± 1.88 | 12.7 ± 1.07 | $P = 0.212$ |
| **Parental care behaviors** | **Group nesting (mean ± S.D.)** | **Solitary nesting (mean ± S.D.)** | **GLMM p-value** |
| Average guarding (%) | 65.46 ± 21.37 | 80.69 ± 30.85 | $P = 0.0004$ |
| Absence (%) | 29.81 ± 16.89 | 19.30 ± 30.8 | $P = 0.00195$ |
| Chasing (%) | 0.04 ± 0.02 | 13.79 ± 6.05 | $P = 0.299$ |
| Oxygenation of progenies (%) | 13.084 ± 9.9 | 12.25 ± 9.57 | $P = 0.358$ |
| **Estimator of fitness** | **Group nesting (mean, range)** | **Solitary nesting (mean, range)** | **GLMM p-value** |
| Total area of received eggs (mm$^2$) | 532 (265–969) | 350 (118–758) | $P = 0.0054$ |
| Hatching success | 0.4 (0.039–0.913) | 0.007 (0-0.033) | $P = 0.00038$ |
| Relative fitness | 1.236 (0.12–4.06) | 0.006 (0.0–0.031) | $P = 0.0442$ |
| Predation (% of area lost) | 60 (9–96) | 99 (97–100) | $P < 0.0001$ |

size (Estimate $= -0.996$, S.E. $= 0.76$, $t$-value $= -1.308$, $p = 0.212$), offspring oxygenation (Estimate $= 0.036$, S.E. $= 0.04$, $t$-value $= 0.91$, $p = 0.358$), nor intruder chasing (Estimate $= 0.591$, S.E. $= 0.569$, z-value $= 1.038$, $p = 0.299$) (Table 2, Fig. 4).

The GLMM $_{EP}$ model to assess whether females might bias their oviposition preferences based on eggs owned by a male in the nest showed significant differences in the receptivity day (Estimate $= 20295$, E.E. $= 3668. 8$, $t$-value $= 5.53$, $p < 0.05$), and the eggs obtained per day per male (Estimate $= 0.725$, E.E. $= 0.156$, $t$-value $= 4.65$, $p < 0.05$), but not of the interaction day*eggs obtained (Estimate $= 0.001$, E.E. $= 0.087$, $t$-value $= 0.012$, $p = 0.9905$). The fact that the interaction is not significant suggests that although there are

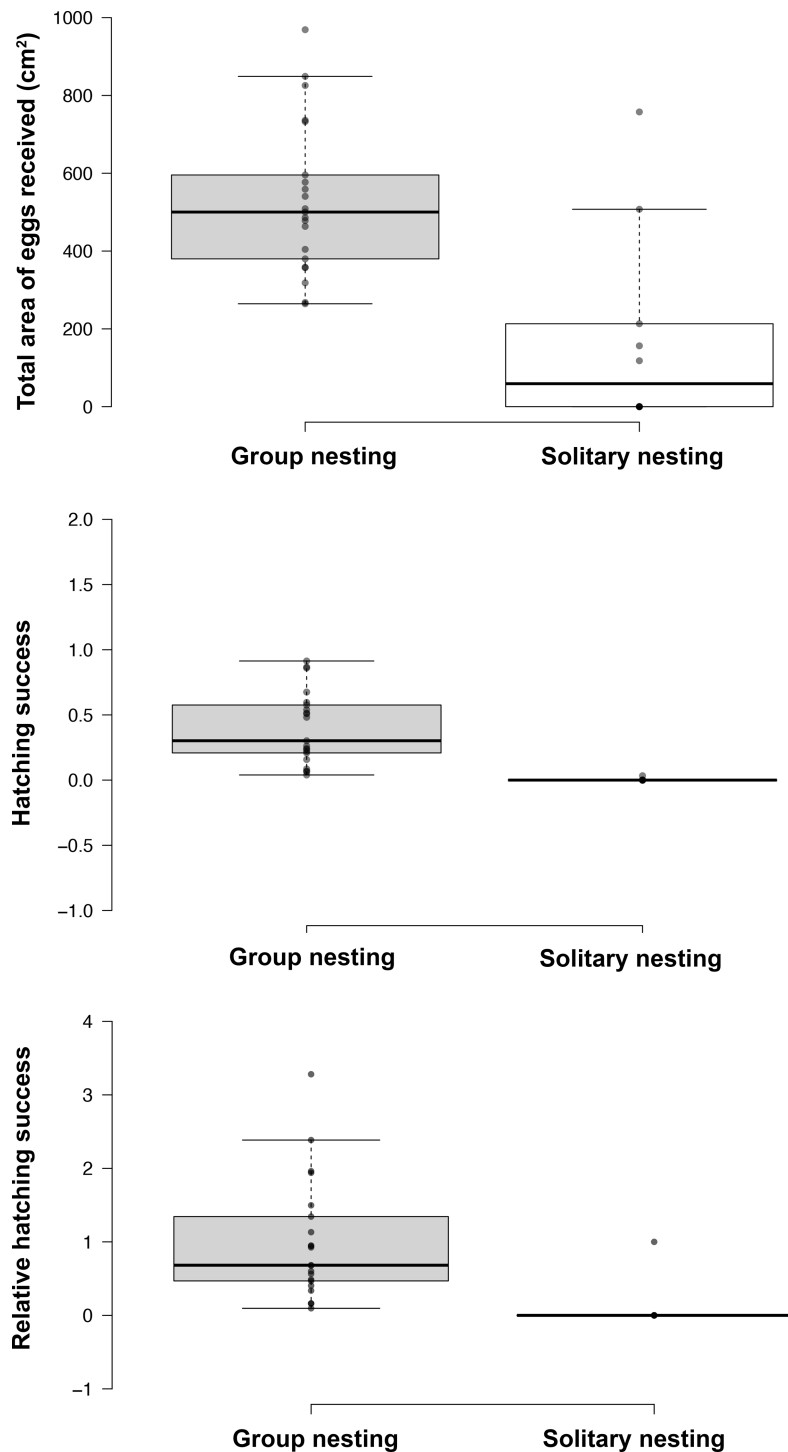

**Figure 3   Effect of the nesting behavior on the total area occupied by oviposited eggs, hatching success and relative hatching success of males of *Abudefduf troschelii*.**  All of the parental care behaviors show significant differences after Bonferroni corrections associated with the spatial distribution patterns of their nests within the colony.

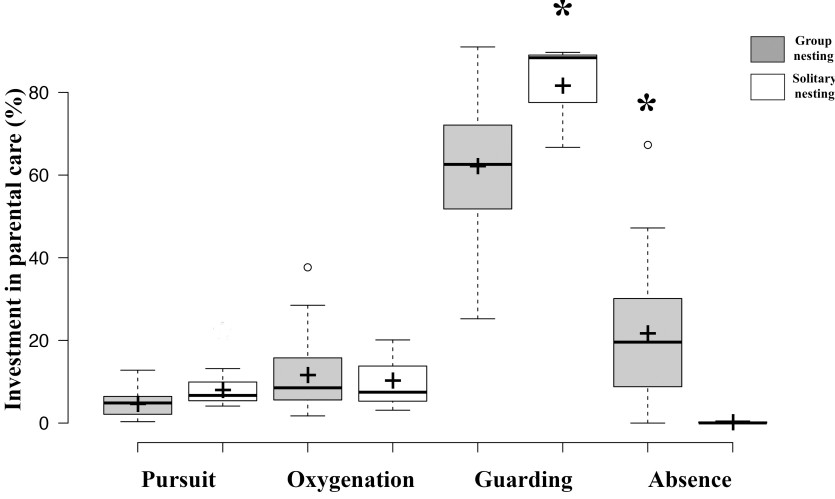

**Figure 4 Investment in parental care behaviors between group nesting and solitary nesting males.** The asterisk (*) shows significant differences between the male nesting strategies for each behavior.

daily differences among males in their mating success, the daily probability of success for each male is not determined by the number of eggs obtained on the previous day (Fig. S3).

The Spearman rank correlation analyses did not indicate any significant relationships between the distances from the center of each territory (Group nesting and solitary nesting males) to the center of the territory of the most successful male in each group and the total length of the males (TL: $S = 6444.1$, $r = -0.298$, $p = 0.102$). However, significant and negative relationships were observed between the distance to the center of the territory of the most successful male in each group and the center of the individual territories of the group nesting and solitary nesting males, for all of the estimators of reproductive success (total area occupied by eggs: $S = 8152.2$, $r = -0.643$, $p < 0.001$; hatching success: $S = 8348.3$, $r = -0.683$, $p < 0.001$; relative hatching success: $S = 8630.4$, $r = -0.739$, $p < 0.001$, Fig. 5). We did not observe a significant relationship between the distances from the center of each territory (Group nesting and solitary nesting) to the centroid of the group and the total length of the males (TL: $S = 5786.1$, $r = -0.166$, $p = 0.3705$). In contrast, significant and negative relationships were observed between the distance to the centroid of the group and the center of the individual territories of the group nesting and solitary nesting males, for all of the estimators of reproductive success (total area occupied by eggs: $S = 7434.2$, $r = -0.498$, $p < 0.001$, hatching success: $S = 8228.1$, $r = -0.659$, $p < 0.001$, relative hatching success: $S = 8149.1$, $r = -0.643$, $p < 0.001$, Fig. S1).

## DISCUSSION

The results of this study regarding the reproductive advantages of the breeding colony mating systems for *A. troschelii* males confirm the theoretical expectation and previous empirical evidence (*Foster, 1989*). Furthermore, our results suggest that, although within the breeding colony, there appears to be a continuous and homogeneous distribution of

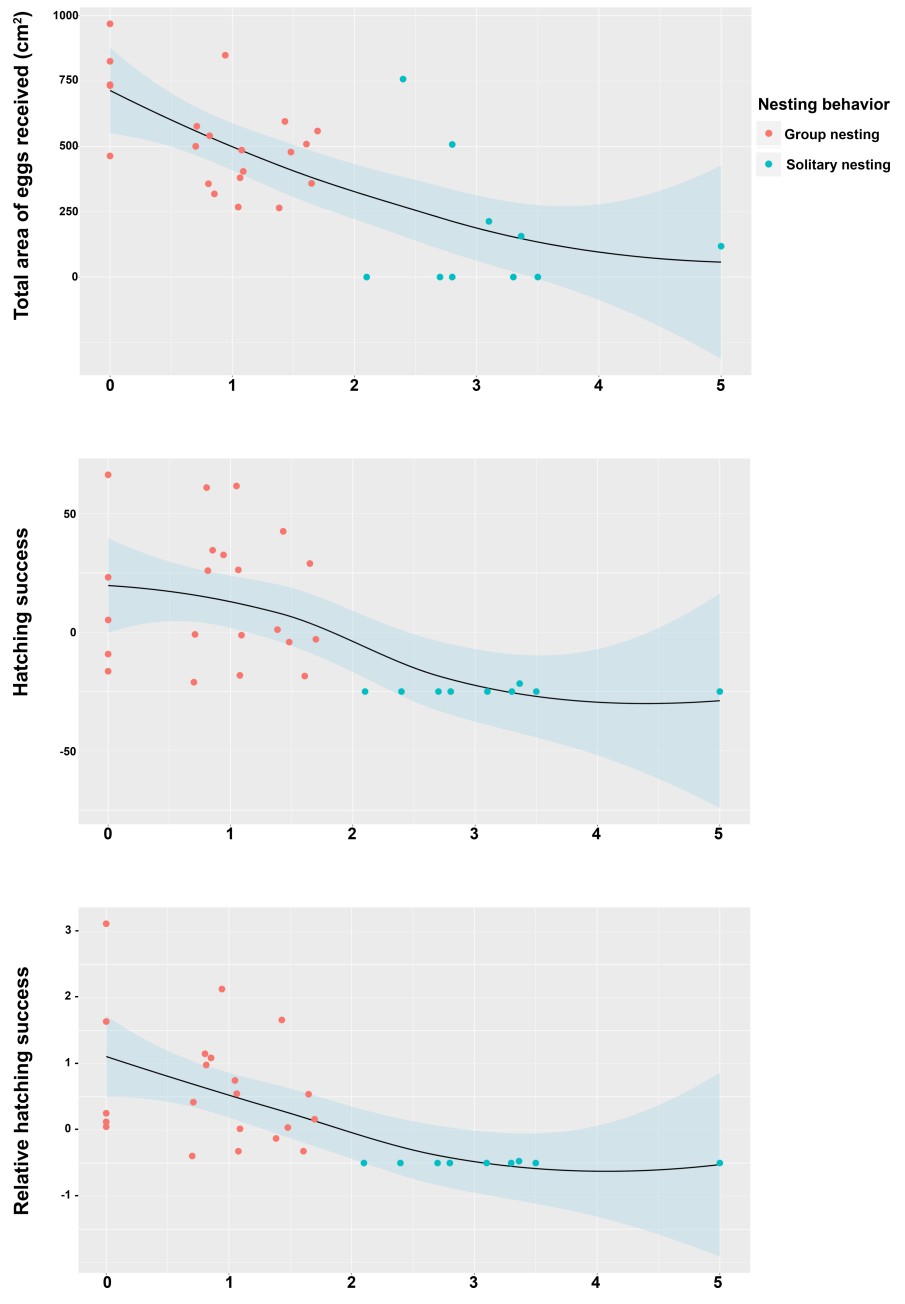

**Figure 5** Relationship between the position of the individual territory of the males to the center of the territory of the most successful male of the group with respect to the estimators of reproductive success. (A) Total area occupied by eggs, (B) hatching success, (C) relative hatching success.

available nesting habitat (although there could undoubtedly be subtle variations in the substrate that we did not measure), *A. troschelii* males do not distribute their territories randomly. Spatial analysis of male territories within this population suggests that larger males perform group nesting behavior, locating their nests closer together than would be expected by chance and these groups are in the central part of the colony. On the other

hand, smaller males perform solitary nesting behavior, locating their territories in the periphery of the colony and far away from the groups and nests of other solitary males.

The effect of male body size and total number of eggs obtained suggests that larger individuals perform group nesting behavior in *A. troschelii*, which may reflect their higher intrasexual competitiveness, defense ability, body condition, or experience (*Coté & Hunte, 1989*; *Maan et al., 2004*; *Dijkstra, Vander Zee & Groothuis, 2008*; *Genner et al., 2008*). Although we do not know the age classes in *A. troschelii*, the average body size difference between group and solitary males (ca. one cm) is consistent with the differences between the two older age classes of *A. saxatilis* males (*Villegas-Hernández et al., 2022*), which along *A. hoefleri* form the sister clade of *A. troschelii*. Supporting that group nesting males are of a different age class (*i.e.,* one year older) than solitary males. Older and more experienced males apparently prefer to establish their nests near other males of the same status. Whereas younger males have to establish solitary territories near these groups as part of their learning process or because they are excluded from the groups.

Besides larger males, only grouping males maintained their nuptial coloration during the 3 days of receptivity. In contrast, only 50% of solitary males adopted nuptial coloration, and just one maintained it until the second day of receptivity. The change in body coloration associated with reproduction (*i.e.,* nuptial coloration) occurs in multiple species of marine and freshwater fishes. It has been described as having functions both in agonistic interactions among males and female attraction (*Kodric-Brown, 1998*). In addition, male coloration ornaments often reflect body condition (*Candolin, 1999*; *Velando, Beamonte-Barrientos & Torres, 2006*), the intensity of parasitism or the condition of the immune response (*Møller, Christe & Lux, 1999*; *Clotfelter, Ardia & McGraw, 2007*), oxidative damage (*Pike et al., 2007*) and territory quality (*Kodric-Brown, 1983*).

Our results show that males with group nesting MS have territories in the center of the colony and closer to the centroid of their group while presenting higher success in all fitness estimators. This oviposition preference for males occupying central territories within a colony has been reported in this same species (*Foster, 1989*). In our study, the relatively small number of males per group makes it difficult to analyze whether the factors determining the fitness advantage of group-nesting males is territory position or simply group membership. However, although we did not measure other phenotypic traits of males, the fact that distance to the centroid is not related to male size could partially support the hotshot model in which the oviposition preference of females is for nests located within the breeding colony and not for male phenotypic traits (*Beehler & Foster, 1988*) nor the presence/number of eggs in the nest.

Similar behaviors have been described in fishes; for example, in cichlids with cooperative colonial breeding males with shorter distances to the nearest neighbor have fitness advantages (*Jungwirth et al., 2015*; *Schütz et al., 2016*). Also, a preference for male settlement within dense colonies has also been experimentally demonstrated (*Heg et al., 2008*). The same advantage of reproductive groups has been described in *Amphiprion frenatus* and *Dascyllus aruanus* (Pomacentridae); where males maintain territories adjacent to that of other breeding pairs and whose group size varies depending on shelter size (*e.g.,* anemone or coral) or food availability (*Williams & Sale, 1981*; *Hattori, 1991*; *Kobayashi & Hattori, 2006*;

*Hattori & Casadevall, 2016*). The influence of group size on male reproductive success has been previously evaluated in *Abudefduf* breeding colonies. However, the effect of solitary nesting has not been analyzed so far, although what we call solitary males have been previously described as "groups of 1 individual" for several species (*Foster, 1989*; *Tyler III, 1992*; *Tyler III, 1995*; *Young et al., 2009*).

Due to our methodology, we cannot affirm that grouped males have higher number of mates. Nonetheless, considering that the average potential fecundity of the females has been estimated to be close to 21,000 oocytes in the ovary per spawning batch of mature females, both for the species *A. saxatilis* (*Villegas-Hernández et al., 2022*) and *A. abdominalis* (Helfrich, 1958); grouped males received at least clutches from 3.4 females per nest, and solitary males on average received eggs from 0.96 mates. Thus, as the female preference model proposed, group-nesting males are preferred (*i.e.,* higher mating success, higher total eggs) because their progeny have a lower probability of predation than in solitary male nests. Similar advantages on the survival of progeny to predation have also been observed in other taxa, such as insects (*Strassmann, Queller & Hughes, 1988*) and birds (*Riehl, 2020*).

Our result is consistent with the reduced predation on eggs of males nesting in larger groups observed in species of *Abudefduf* (*Foster, 1989*; *Tyler III, 1995*). Although, it is possible that some of this predation could be associated with filial cannibalism by the male, as has been reported in species such as *A. luridus* (*Afonso & Santos, 2005*) and *A. sexfasciatus* where about 13% has been reported (*Manica, 2003*). However, during the evaluation of paternal care behaviors, we did not observe any filial cannibalism behavior (*i.e.,* nips to eggs), which coincides with the results of previous work in the same population (*Pérez-Hernández, 2018*), suggesting that the frequency of filial cannibalism in this population of *A. troschelii* is relatively low. Considering the level of predation is essential because some solitary nesting males obtained eggs at the receptivity stage, but almost all (*i.e.,* 89%) lost them before hatching. The average percentage of oviposited eggs that reached the end of the nesting phase was only 1% for the solitary nesting males, compared to 40% for the Group nesting males. It has been reported for this same species that the males with closer neighbors have greater success in the eclosion of larvae than those for which the nearest neighbors are located at longer distances (<1 m; *Foster, 1989*).

Because of the high predation pressure that colonies suffer, only group defense may guarantee egg hatching. However, solitary nesting at a breeding colony may be maintained in the population if solitary males exploit the efforts of group males by establishing close territories within the breeding colony. Locating their territories close to the groups allows them to act opportunistically by intercepting females attracted to the colony and obtaining some egg clutches or even stealing the paternity of some group nests, which according to previous results in this population of *A. troschelii* could represent the potential of sneaker males to sire up to 50% of the clutches (*Guillen-Parra et al., 2020*). Furthermore, both could be sequential strategies associated with changes in ontogeny (*i.e.,* size) or changes in the status of individuals that could be maintained in the population as an optimal response (*Brockman & Taborsky, 2008*).

It is notable that, contrary to that described by *Tyler III (1995)* in *A. Abdominalis*, the *A. troschelii* males with the solitary nesting lost almost all of the eggs they received. The

males of the study population nest on an apparently substrate continuum and are thus probably just as visible and possibly detectable by the predators as the grouped males, and the additive defense of group nesting seems to provide a significant advantage in reducing predation. In order to determine whether it is the female preference or hotshot model that best explains the formation of the Nesting groups, follow-up or experimental manipulations are required in order to observe whether it is the successful males that attract the females to the Nesting group or whether it is the size of the group itself that is the attractant (*Alatalo et al., 1992*).

The GLMM results indicate that male size and parental care investment are important factors in explaining differences in reproductive success and predation reduction between group-nesting and solitary-nesting males. Our results strongly suggest that group nesting is advantageous for *A. troschelii* males in addition to advantages in reproductive success because it markedly reduces the cost of investment in parental care behaviors. We observed a reduction in the time allocated to guarding behaviors, as well as in the time and number of aggressive interactions against intruders. Similar results have been reported in cooperatively breeding colonies of cichlids, with larger groups reducing their investment in anti-predator behaviors (*Jungwirth et al., 2015*). It remains to be determined whether this reduction in the energy budget associated with the cost of paternal care allows individuals with group nesting behavior to maximize their fitness by allowing them increased participation in successive reproductive events. Spatial patterns of nesting show that the proximity of nests to the center of groups confers significant fitness advantages on males. However, there is a need to determine the patterns that govern group establishment (*e.g.*, phenotype matching; also whether these groups maintain their stability over time, and the mechanisms that may shape the patterns of association within these groups (*e.g.*, familiarity), as well as to deepen our understanding of female choice mechanisms.

The documentation of alternative reproductive tactics, such as those displayed by the males of this population, has been reported in multiple species that exhibit territorial and parental care behaviors (*e.g.*, in families such as Cichlidae, Labridae, Gasterosteidae and Pomacentridae). The section above, supports the theoretical prediction that exclusive paternal care of progeny can be exploited by competing males and promotes the evolution of alternative reproductive tactics (ARTs, *Taborsky, 2008*; *Oliveira, Taborsky & Brockmann, 2008*).

The two nesting strategies described in this study (group nesting and solitary nesting) involve territorial defense and parental care of the progeny, and have been suggested indirectly in *Abudefduf saxatilis*, *Abudefduf troschelii* (*Foster, 1989*) and *Abudefduf abdominalis* (*Tyler III, 1995*), although the previous operational definition of "groups of males" in this genus does include groups formed by a single individual (*Tyler III, 1995*), which corresponds to the definition used in this study of the solitary nesting. This study is therefore the first documentation of two territorial strategies in the same population of pomacentrid fish with a breeding colony mating system, as well as the differential parental investment associated with each of these strategies. In this same context, the wide variance in reproductive success among the males of the study population seems to promote the appearance of alternative reproductive tactics (*Mendoza-Cuenca & Macías-Ordóñez, 2010*),

and may even explain the existence of a third "sneaking" ART previously suggested in the males of *A. troschelii* in this study population (*Guillen-Parra et al., 2020*) and for *A. sordidus* (*Lobel et al., 2019*). Even though our study focuses on the analysis of a reproductive cycle in a population of *A. troschelii* present in the Sea of Cortez, The similarities with results previously reported in *A. troschelii* in Panama (*Foster, 1989*) and the important homogeneity in space and time of marine regions such as the southern Gulf of California and the Mexican Pacific Transition, could imply that our results could be repeatable or common in *A. troschelii* populations distributed throughout these biogeographic regions.

Similar behavior has been reported in the salamander *Hemidactylium scutatum* in which the females can present one of three ARTs: solitary nesting with care of the progeny, group nesting of females with care of the progeny and oviposition of eggs in the nests of other females (*Harris et al., 1995*). Since these two nesting strategies have not been previously reported, we cannot conduct a direct comparison with the data of other studies; however, *Tyler III (1995)* found that *A. abdominalis* males in larger groups (*i.e.,* more than 15 individual) presented greater mating success than those in smaller groups, since the definition of group provided by that author included solitary nests, with small groups defined as those formed by one to seven individuals. It is possible that those solitary individuals also had lower fitness than those in groups, but this pattern could not be observed when defining groups of one individual.

## CONCLUSIONS

The colonial nesting mating system of *A. troschelii* males provides asymmetrical benefits to individuals. *A. troschelli* males non-randomly distribute their nests within the breeding colony. Larger males establish nesting groups within the colony, while smaller males establish solitary territories on the periphery of the colony. Females present a bias in their mating preferences towards group-nesting males, which may be associated with these males having greater intrasexual competitiveness, defensive ability, body condition, and experience or because their eggs will have a lower probability of predation when in grouped nests. In this context, females may be assessing the quality of males through the presence, intensity, and duration of nuptial coloration presented by males. The reduction in male parental investment in paternal care behaviors is remarkable, and it remains to be determined whether "time saved" allows males to have more nesting events throughout the year. The observed nesting behaviors could represent two alternative mating tactics. The solitary nesting tactic may allow paternity theft from males with group nesting tactics.

## ACKNOWLEDGEMENTS

We thank Michael Taborsky and the four anonymous reviewers for their constructive comments that highly improve the quality of the manuscript, and special thanks to Alicia Chávez, Mauricio Guillen & Francisco Martínez for their help in the fieldwork.

### Funding

Mariana Solís-Mendoza was supported by the Consejo Nacional de Ciencia y Tecnologia scholarship granted during their Ph.D. studies (CONACyT 743323). The División de Estudios de Posgrado de la Universidad Michoacana de San Nicolás de Hidalgo provided funds for the proofreading of the English version of the article. The funders had no role in study design, data collection and analysis, decision to publish, or preparation of the manuscript.

### Grant Disclosures

The following grant information was disclosed by the authors:
Consejo Nacional de Ciencia y Tecnologia: CONACyT 743323.
División de Estudios de Posgrado de la Universidad Michoacana de San Nicolás de Hidalgo.

### Competing Interests

The authors declare there are no competing interests.

### Author Contributions

- Mariana Solís-Mendoza conceived and designed the experiments, performed the experiments, analyzed the data, prepared figures and/or tables, authored or reviewed drafts of the article, and approved the final draft.
- Omar Chassin-Noria conceived and designed the experiments, performed the experiments, analyzed the data, prepared figures and/or tables, authored or reviewed drafts of the article, and approved the final draft.
- Carlos Levi Pérez Hernández conceived and designed the experiments, performed the experiments, analyzed the data, authored or reviewed drafts of the article, and approved the final draft.
- Luis Mendoza-Cuenca conceived and designed the experiments, performed the experiments, analyzed the data, prepared figures and/or tables, authored or reviewed drafts of the article, and approved the final draft.

### Data Availability

The data that supports the findings of this study are available at OSF: Mendoza-Cuenca, Luis. 2023. "Reproductive Benefits and Reduced Investment in Parental Care Behavior Associated with Reproductive Groups of Males in Abudefduf Troschelii." OSF. July 9. doi: http://dx.doi.org/10.17605/OSF.IO/KZR73.

### Supplemental Information

Supplemental information for this article can be found online at http://dx.doi.org/10.7717/peerj.15804#supplemental-information.

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
