# Peer review of "Reproductive benefits and reduced investment in parental care behavior associated with reproductive groups of males in Abudefduf troschelii"

_PeerJ, doi:10.7717/peerj.15804_

## Round 0.1 · original submission · Major Revisions

Most of the comments of reviewers suggest numerous changes to the manuscript. Hence, it is recommended to revise the manuscript.

Reviewer 1 ·

Basic reporting

After reviewing the manuscript entitled "Reproductive benefits and reduced investment in parental care behavior associated with reproductive groups of males in Abudefduf troschelii" This manuscript represents a contribution to the reproductive biology of reef fish, mainly in species in which the males care for their offspring. It is well-written, the objectives are clear, the materials and methods are well-detailed. The statistical analysis is considered adequate according to the type of data that the authors generated, the results and the discussion are well-described and well-founded respectively. I only have an observation and suggestion for the authors in the parts where they mention pursuit change to chasing since it is more appropriate. Therefore I believe that the article should be accepted for publication with that small change.

Experimental design

It is already answered in the previous section

Validity of the findings

It is already answered in the previous section

Additional comments

No comments

Reviewer 2 ·

Basic reporting

The ms is very well written. The theme is super important for nesting ecology. The references used were well chosen. The experimental design is correct. The analyzes chosen are satisfactory. However I suggest only two very important things:
-please insert the data points (all data, using little dots, jitter plots) in the box plot figure (fig 3 and 4), not just the outliers.
-In the figure with the models (fig 5) it is very important to insert the confidence intervals;
-In the figure 5, since you have group and solitary nest category, why not two models, instead only one?

Experimental design

Totally ok

Validity of the findings

no comment

Additional comments

Congratulation for the important research

Reviewer 3 ·

Basic reporting

Use of English
While the standard of English is good for the most part, there are a number of places in the manuscript where issues with phrasing hampers the meaning. These are:
Line 50 Change “;” to “,”
Line 57 “or in solitary”: phrasing is not correct
Lines 65-66 The meaning of “while allowing group-nesting males to reduce their parental investment in offspring’s paternal care behaviors” is unclear.
Line 130 “engage in” rather than “present”
Lines 197-199 Not a proper sentence – meaning unclear
Lines 249-250 Not a proper sentence
Lines 280-281 Not a proper sentence
Lines 367-373 Issues with phrasing in this section
Line 405 “Groups of ‘1 individual’” is a confusing phrase – can’t these just be referred to as solitary males?
Lines 443-445 The phrasing here is confusing; in particular, it is difficult to keep track of which kind of male is being discussed.
Line 459 “Reproductive alternatives” as a phrase is rather awkward.

Context and background literature
Most literature relevant to the system and questions appears to have been captured, but there are a couple of gaps:
Line 243 Is it known whether males of this species engage in filial cannibalism as at least one other species in this genus (A. sexfasciatus) does? If some males eat a proportion of their eggs, then the area of eggs recorded is unlikely to correspond to the total area of eggs received. Furthermore, in the Discussion, the authors refer a number of times to egg predation by heterospecifics and argue that minimising the risk of predation might in part select for nesting in close proximity to other males. To assess this idea, it is crucial to know whether the disappearance of eggs is indeed due largely to predation from heterospecifics or whether males could be eating substantial numbers of their own eggs.
Lines 275-276 It would be helpful for the authors to note at this point that such a preference is indeed seen in another Abudefduf species (sexfasciatus), and in teleost species with male-only parental care more widely (Goldberg et al. 2020 Proceedings of the Royal Society B).
As a more general comment, I'd encourage the authors to look beyond fish to work on birds and insects that demonstrates clear benefits to cooperative/communal breeding in terms of reducing the risk of offspring predation. The results presented in the manuscript suggesting the males may benefit from nesting together despite intense competition for mates provides a fascinating point of comparison to these systems where predation risk drives females to collaborate in nesting despite significant costs of reproductive competition. In particular, work on anis by Riehl and paper wasps by e.g. Strassmann would be worth looking at in relation to the results for the damselfish and some comparison with these systems would help to set the results in a broader context.

Article structure
The article is sensibly structured and the inclusion of the figures and tables is appropriate. I could not, however, see any reference to raw data and have not been able to access them to check that they are in a suitable format.

Results relevant to hypotheses
The results presented in the manuscript are relevant to the stated aims of the study.

Experimental design

The research presented in the manuscript is original and falls within the Aims and Scope of the journal.

The main research questions could be stated more clearly - while the rationale underpinning the individual tests that the authors perform is generally clear, a bit more work is needed to emphasise the key aims of the study in the Introduction. For instance, at Lines 100-102 it's implied that an aim of the study is to understand “the selective patterns that determine the formation and structure of the reproductive group” but it's not clear from this wording what phenomenon the authors are really describing.

I have no concerns about the rigour with which the authors carried out the work - all field methods seem appropriate and carried out to a high standard. While data collection is for the most part clearly described, there are some places where clarification or additional information is needed:
Lines 205-206 More detail is needed when describing the blinding procedure. I assume that observers were blind to the status of males as group-living or solitary, but this needs to be made clear.
Lines 211-212 Are there particular postures associated with guarding behaviour that would indicate the fish is monitoring the environment? Such information would help to improve reproducibility of the behavioural scoring method.
Lines 207-214 It seems strange to classify “absence” as a paternal care behaviour – I suggest the authors limit the categories of paternal care to chasing, fanning and guarding, and then note that in addition to these behaviours the male was sometimes absent.

In terms of animal welfare, my own assessment is that the methods used are unlikely to have caused significant or lasting stress to the fish. However, it is troubling that no reference is made to any ethical review or oversight of the work - if the authors sought ethical approval, this should be made clear. If not, it would come down to the journal's policy as to whether the work can be published.

Validity of the findings

My main concerns with this manuscript relate to the approach taken to data analysis. Although the authors have clearly gone to a great deal of trouble to collect high-quality data from the field (admirable given the difficulties of studying fish in the field), I nonetheless have serious reservations about the analyses used, to the extent that I would not have full confidence in the results as currently presented.

(a) In the GLM analyses, males from the same nesting group are treated as independent data points, but it seems unlikely that they are in fact independent, since they will experience similar conditions which may cause their reproductive success to differ from males in other groups and from solitary-nesting males. I would therefore strongly recommend that the authors rerun these analyses using mixed-effects models with a random term accounting for group identity.
(b) A Poisson distribution is appropriate for modelling count data, but in the GLMs the response variables (total area of eggs, area of hatched eggs/total area of eggs and area of hatched eggs/mean total area across all males) are continuous, non-integer values. These analyses need to be rerun using an appropriate distribution – (mixed-effects) linear models with a Normal distribution may be a good choice here.
(c) Following on from the above point, I think that the results for the GLMTA analysis presented in Table 1 clearly point to an issue with this analysis. For all of explanatory variables bar “absence”, the z values are enormous, which suggests to me that the data are not a good fit to the Poisson distribution that’s being used.
(d) I don’t understand the rationale for excluding males that fail to receive eggs. Surely by doing so the authors are underestimating the variance in reproductive success among males, something which is implied to be a central aim of their study? If the authors are concerned about the inclusion of zero values, there are statistical approaches that can allow modelling of continuous zero-inflated data, for example compound poisson GLMMs.
(e) In addition to the GLM analyses of reproductive success, the authors compare male size, behaviour and reproductive success (again) between group-living and solitary-nesting males using Mann-Whitney tests (although there is confusion about the exact test they used – see next point). Again, these tests assume that males within an individual breeding group are independent, but as explained above this is unlikely to be true. I would therefore advise the authors to instead use mixed-effects models to compare male size and behaviour measures between group-living and solitary males with group ID as a random effect (i.e. the same approach recommended for analysing reproductive success measures in (a) above).
(f) As indicated above, there is confusion about the type of test used to compare characteristics of group-nesting and solitary nesting males. Individuals in these two groups are not in any sense paired, and so the analysis is simply comparing the medians of the two groups. I therefore assumed that the authors used a Mann-Whitney test, which does not require paired observations. This appears to be supported by the phrasing at Line 272, although the reference here to “Wilcoxon Mann Whitney” is confusing, since Wilcoxon sign tests use paired data. However, at Lines 325-326 in the Results the authors refer to a “Wilcoxon paired sample test”, which is not the same as a Mann Whitney test and would not be appropriate for comparing group-living and solitary males (although as I explain above I believe the authors ought really to replace these analyses with mixed-effects models to deal with the nonindependence within breeding groups).
(g) I’m unclear as to how the analyses described at Lines 277-279 allows the authors to test for the effect they are interested in. As I understand it, the aim here is to test whether females prefer to spawn with males whose nests contain a larger number of eggs. I don’t see how this is addressed by this analysis, which asks whether the number of eggs received on a given day changes over the period of receptivity. Surely a significant effect of day of receptivity would be observed simply as a result of a female choosing to spawn with a male who up to that point had no eggs? I think a better test would be to ask whether the number of additional eggs received on a given day is predicted by the number of eggs present on the previous day – is this feasible?
(h) The authors used Spearman rank correlation tests were used to explore associations between distance measures and reproductive success. I would expect the output of such tests to include the correlation coefficient (r) and the p value but in the Results the authors quote F values. How were these obtained, and how do they relate to r values?
(i) Line 298. Bonferroni-adjusted P values are routinely applied as a correction for multiple comparisons but I’m not aware of them being used as a penalty for nonindependence in data (which I assume is what’s meant by “no[n]independence between variables”. Rather, steps should be taken to handle the nonindependence in the analysis itself, for instance by using random effects or using simulations (see suggestions above).
(j) Finally, there are a number of important details missing from the description of the analyses:
(1) How did the authors check for overdispersion in models with a Poisson distribution?
(2) How were z and P values obtained for explanatory variables in the GLM and LMM analyses? Were these taken from the global model containing all variables or were models simplified so that only significant terms were retained in a minimum adequate model. Making this explicit is crucial if readers are to be able to interpret model outputs correctly.
(3) Were all tests two-tailed?
(4) When Bonferroni corrections were applied, what cut-off was used for statistical significance? In all other cases, did the authors use a cut-off of 0.05?

In addition to concerns about the analyses, the presentation of the results is also problematic. At Lines 314-334, where results of the GLM and Mann Whitney tests are presented, it is not possible to discern what the effects actually are and I would strongly urge the authors to rewrite this section. Rather than state “there was an effect of variable X” in the GLM analyses, please tell us what the effect was e.g. “total area of eggs received increased with X”. Similarly, rather than state “there was a difference between group and solitary-nesting males” for the Mann-Whitney comparisons, tell us what the direction of this difference was e.g. “hatching success was significantly higher in group-nesting versus solitary males”.

The other issue with the presentation of results concerns Figure 5. What do the lines on the plots represent? They seem to indicate nonlinear relationships between distance and hatching success measures, but such relationships were not tested for in the analyses. If these are lines of best fit, then I’d also expect to see confidence intervals around the lines in each case.

In terms of conclusions, these were generally appropriate given the results that the authors presented, but as discussed above these may not be robust given the choice of analyses. In a couple of places, however, I think it would be helpful for the authors to think again about what they are able to conclude from their data:
Lines 360-362 While the results show that males’ nests are not distributed randomly within the habitat, I don’t think the authors actually demonstrate that there is a “continuous distribution of available habitat”. To do so would require showing that males could have successfully nested in unoccupied sites – although these might appear similar to chosen sites, there may be subtle differences in e.g. shelter or substrate that make these unchosen sites unsuitable.
Lines 391-394 Given that no other phenotypic traits were measured, I would argue the suggestion that females are choosing males based on nest sites rather than phenotypic characteristics is a bit too strong.

Additional comments

Lines 104-106 The indirect fitness benefits of lekking need to be clarified – accrual of indirect fitness depends on cooperative interactions with kin but a description of such interactions is missing.
Lines 125-126 The current wording implies that the distance between nests influences the likelihood that individuals will engage in ARTs. Is this correct? A more general statement would be that the risk of paternity loss increases with decreasing distance between nests, something that presumably holds true for species with and without ARTs.
Lines 183-184 The authors state that group-nesting males are found in groups of 3 or more with <1m to the nearest neighbour’s nest, while solitary males nest >2.5m from their nearest neighbour’s nest. Does this imply that two males were never observed to nest within 1m and at a distance of >2.5m from other males?
Line 265 There is some confusion with the terms used to indicate the start of the observation period. Earlier in the manuscript, the authors state that data were collected from the onset of spawning but here they state that the day of nest cleaning was used. It would be helpful if the authors could clarify the temporal relationship between nest cleaning and spawning.
Lines 335-338 Please see my comments above – I’m not convinced that this analysis is the best way to determine whether females show a preference for males with (lots of) eggs.
Lines 358-360 Rather than stating simply that the results match theoretical predictions and previous work, the authors need to explain how their results support earlier work.
Line 407 By “mating success”, do the authors mean number of mates? It would be helpful to clarify that here.
Lines 456 Phenotype matching is a mechanism of kin recognition that allows recognition of unfamiliar relatives through comparison to one’s own phenotype or that of a known relative. In the current phrasing, however, it’s implied that phenotype matching and information about kinship are two distinct processes that could shape group formation.

Reviewer 4 ·

Basic reporting

This manuscript deals with male reproductive strategies in Abudefduf troschellii.
The paper is very well written. It places this study in a much broader context and draws some significant and encompassing conclusions.
Experimental design and statistical analyses are very nicely done, conclusions are cogent and well supported, the discussion is interesting.

Overall, I find this paper excellent and worthy of publication.

Experimental design

no comment

Validity of the findings

the paper is well written so my comments are of the 'minor comments' category

1. it is stated that saxatilis is sister to the focal species. it is, sort of. If one wants to be completely precise, saxatilis is sister to hoefleri, and that clade is sister to troschelii.

2. The authors should state in the discussion that their study is likely to be general to the species but it is limited in space (Sea of Cortez) and time (one reproductive cycle). Thankfully, the TEP is very homogeneous in space and time and these results are very likely to be common throughout the biogeographic region.

3. the discussion about fitness, the authors do not mention what seems the most important thing to me, which is the potential for sneaker males to sire 50% of the cluthes, this means that fitness is very difficult to evaluate using behavioral studies.

Additional comments

As I said above, I find this paper to be excellent and a wonderful and interesting read.

---

## Round 0.2 · Minor Revisions

There are still some minor corrections, kindly address the issues and submit again

Reviewer 1 ·

Basic reporting

After reviewing the corrections made by the authors to the suggestions made by the reviewers to the manuscript entitled "Reproductive benefits and reduced investment in parental care behavior associated with reproductive groups of males in Abudefduf troschelii." I consider that they were carried out adequately, which allowed us to improve this manuscript. Therefore, I recommend that this be accepted for publication in the Journal in its current state.

Experimental design

.

Validity of the findings

.

Reviewer 3 ·

Basic reporting

At Lines 342 to 350 in the revised manuscript, the authors describe results from some of their new mixed-effects models. The quality of English in this paragraph is uneven and should be checked.

Experimental design

I continue to disagree with the authors about the logic of classifying male absence as a form of parental care. In their replies, the authors suggest (I think) that this could constitute care if, by not attending the nest, the male is unable to eat the brood. This seems highly unlikely, and impossible to test; much more likely, the male leaves the brood to forage, in which case his absence is linked to maintaining his own condition rather than directly caring for the offspring. I would again urge the authors to remove the father's absence from their list of care behaviours, but if they choose to keep it in they should provide a clear reason for doing so in the manuscript (i.e. explain why it is a form of parental care).

Validity of the findings

At Lines 345-349 in the revised manuscript, the authors present results for guarding traits and percentage eggs lost. The results are identical (same estimates, standard errors, t values and p values) - I'd encourage the authors to check to see whether this is a mistake.

Additional comments

I am very grateful to the authors for their positive engagement with my comments and for the work they put into revising the manuscript, which I enjoyed reading. I only have a few tiny comments for them to consider.

Reviewer 4 ·

Basic reporting

The authors have satisfactorily addressed all the reviewers concerns and in my opinion this paper is now ready for publication as is.

Experimental design

NA

Validity of the findings

NA

Additional comments

NA

---

## Round 0.3 · accepted · Accept

All the reviewers are in favour of the manuscript, hence, my recommendation is to accept the manuscript.

Reviewer 1 ·

Basic reporting

After reviewing the corrections made by the authors of the publication entitled "Reproductive benefits and reduced investment in parental care behavior associated with reproductive groups of males in Abudefduf troschelii" to the suggestions of the reviewers, these were made satisfactorily, which allowed substantially improve the manuscript, so I recommend that it be accepted for publication in its current state.

Experimental design

See above

Validity of the findings

See above

Additional comments

No comments

Reviewer 2 ·

Basic reporting

The authors did an excellent job, thus I suggest publishing the study.

Experimental design

See above

Validity of the findings

See above

Additional comments

The authors did an excellent job, thus I suggest publishing the study.

Reviewer 3 ·

Basic reporting

After two rounds of review, all my concerns have now been addressed and I would be happy to see this manuscript published.

Experimental design

See above

Validity of the findings

See above

Reviewer 4 ·

Basic reporting

The authors have satisfied all my comments and requests.
I think that this new version is an enjoyable read, and an important contribution to the field.
It is ready for publication

Experimental design

NA

Validity of the findings

NA

Additional comments

None